# Urolithin A Protects Hepatocytes from Palmitic Acid-Induced ER Stress by Regulating Calcium Homeostasis in the MAM

**DOI:** 10.3390/biom14121505

**Published:** 2024-11-26

**Authors:** Gayoung Ryu, Minjeong Ko, Sooyeon Lee, Se In Park, Jin-Woong Choi, Ju Yeon Lee, Jin Young Kim, Ho Jeong Kwon

**Affiliations:** 1Chemical Genomics Leader Research Laboratory, Department of Biotechnology, College of Life Science and Biotechnology, Yonsei University, Seoul 03722, Republic of Korea; murphy970928@gmail.com (G.R.); komj0714@yonsei.ac.kr (M.K.); leesy9967@yonsei.ac.kr (S.L.); tpdls0303@yonsei.ac.kr (S.I.P.); 2Digital Omics Research Center, Korea Basic Science Institute, Ochang 28119, Republic of Korea; realbear93@gmail.com (J.-W.C.); jylee@kbsi.re.kr (J.Y.L.); jinyoung@kbsi.re.kr (J.Y.K.); 3Critical Diseases Diagnostics Convergence Research Center, Korea Research Institute of Bioscience and Biotechnology, Daejeon 34141, Republic of Korea

**Keywords:** Urolithin A, ER stress, MAM, CETSA-LC-MS/MS, SERCA, MAFLD

## Abstract

An ellagitannin-derived metabolite, Urolithin A (UA), has emerged as a potential therapeutic agent for metabolic disorders due to its antioxidant, anti-inflammatory, and mitochondrial function-improving properties, but its efficacy in protecting against ER stress remains underexplored. The endoplasmic reticulum (ER) is a cellular organelle involved in protein folding, lipid synthesis, and calcium regulation. Perturbations in these functions can lead to ER stress, which contributes to the development and progression of metabolic disorders such as metabolic-associated fatty liver disease (MAFLD). In this study, we identified a novel target protein of UA and elucidated its mechanism for alleviating palmitic acid (PA)-induced ER stress. Cellular thermal shift assay (CETSA)-LC-MS/MS analysis revealed that UA binds directly to the sarcoplasmic/endoplasmic reticulum Ca^2+^-ATPase (SERCA), an important regulator of calcium homeostasis in mitochondria-associated ER membranes (MAMs). As an agonist of SERCA, UA attenuates abnormal calcium fluctuations and ER stress in PA-treated liver cells, thereby contributing to cell survival. The lack of UA activity in *SERCA* knockdown cells suggests that UA regulates cellular homeostasis through its interaction with SERCA. Collectively, our results demonstrate that UA protects against PA-induced ER stress and enhances cell survival by regulating calcium homeostasis in MAMs through SERCA. This study highlights the potential of UA as a therapeutic agent for metabolic disorders associated with ER stress.

## 1. Introduction

Metabolic disorders such as metabolic-associated fatty liver disease (MAFLD) [1], type 2 diabetes, and obesity are major health problems worldwide. One of the key factors in the development and progression of these metabolic disorders is endoplasmic reticulum (ER) stress. ER stress occurs when the function of the ER is disrupted by protein misfolding or calcium imbalance [2]. This stress triggers a series of responses known as the unfolded protein response (UPR) to restore ER function [3]. However, if ER stress persists and becomes severe, it can lead to cellular dysfunction.

Several proteins and small molecules have been investigated for their role in regulating ER stress, including agents that promote protein folding, maintain ER calcium homeostasis, activate UPR pathways, and alleviate oxidative stress [4,5,6]. In particular, naturally derived compounds such as Urolithin A (UA), a metabolite of ellagitannin, the flavonoid Quercetin, and the polyphenol Resveratrol have been shown to alleviate ER stress [7,8,9]. A previous study showed that UA inhibits ER stress and inflammatory responses induced by glucose toxicity in pancreatic β-cells [7]. UA is a well-known mitophagy inducer that removes damaged mitochondria and also has significant antioxidant and anti-inflammatory effects [10,11,12]. Because of these activities, UA holds great promise as a natural treatment for metabolic diseases. However, its target proteins and detailed mechanisms of action remain to be fully elucidated.

In this study, we investigated the potential of UA to alleviate ER stress associated with MAFLD. Recently, MAFLD has been highlighted as a metabolic disease that includes fatty liver disease and is closely related to insulin resistance and obesity [13]. Using a cellular thermal shift assay (CETSA) [14,15,16,17], we identified sarcoplasmic/endoplasmic reticulum Ca^2+^-ATPase (SERCA) as a novel binding protein of UA.

SERCA uses energy from ATP hydrolysis to pump cytosolic calcium ions into the ER or SR [18]. Dysfunction of SERCA is associated with diseases such as muscle disorders and heart failure [19,20]. The development of modulators of SERCA may be a promising strategy to maintain cellular homeostasis due to SERCA dysfunction. Our results show that UA, as an agonist of SERCA, alleviates palmitic acid-induced ER stress in liver cells and protects cells from ER stress. This study provides a new understanding of how UA regulates ER stress and highlights its potential role in the treatment of metabolic disorders.

## 2. Methods and Materials

### 2.1. Cell Culture and Treatment

HepG2 and HEK293 cells were grown in DMEM, supplemented with 10% FBS and 1% antibiotics. All cell cultures were maintained at pH 7.4 in a humidified incubator at 37 °C under 5% CO_2_ in air. When cells were treated with PA, DMEM containing 2% BSA was used.

### 2.2. Western Blot

Soluble proteins were harvested from cells by using SDS lysis buffer (50 mM Tris-HCl, pH 6.8 containing 10% glycerol, 2% SDS, 10 mM dithiothreitol, 0.005% bromophenol blue). Equal amounts of proteins were analyzed by 8%, 10%, and 12.5% sodium dodecyl sulfate–polyacrylamide gel electrophoresis (SDS-PAGE) and then transferred to polyvinylidene difluoride (PVDF) membranes. The blots were then blocked and immunolabeled with primary antibodies against GRP78/BiP (Proteintech, Chicago, IL, USA, 66574-1-Ig), PERK (Cell Signaling Technology, Danvers, MA, USA, C33E10), p-PERK (Abclonal, Woburn, MA, USA, AP1501), eiF2α (Cell Signaling Technology, Danvers, MA, USA, L57A5), p-eiF2α (Cell Signaling Technology, Danvers, MA, USA, D9G8), CHOP (Proteintech, Chicago, IL, USA, 15204-1-AP), β-tubulin (Abcam, Cambridge, UK, ab6040), β-actin (Abcam, Cambridge, UK, ab6276), SERCA (Proteintech, Chicago, IL, USA, 67248-1-Ig), and Flag (Proteintech, Chicago, IL, USA, 20543-1-AP) overnight at 4 °C. Immunolabeling was visualized using an enhanced chemiluminescence kit (Bio-Rad Laboratories, Hercules, CA, USA, 170-5061) according to the manufacturer’s instructions. Images were quantified using Image J software (ver. 1.53) (National Institutes of Health, Bethesda, MD, USA). All band intensities are proportional to the amount of the target protein on the membrane within the linear range of detection. Western blot original images can be found in Appendix A.

### 2.3. CETSA

Cell suspensions at a concentration of 300 × 10^4^ cells/15 mL were aliquoted into each conical tube. DMSO (control) or UA was treated at 37 °C for 1 h with gentle mixing. After centrifugation, the pellet was washed with PBS and resuspended in 1 mL of PBS (with protease inhibitor), aliquoted into PCR tubes (100 μL per tube), and heated between 40 and 64 °C for 3 min and cooled at 25 °C for another 3 min in the thermal cycler. The tubes were centrifuged to pellet the cells, and the supernatant was discarded and replaced with 0.4% NP-40 (in PBS) supplemented with protease inhibitors to facilitate the solubilization of hydrophobic proteins. The cell suspension was subjected to two freeze–thaw cycles in liquid nitrogen and centrifuged at 20,000× *g* for 20 min at 4 °C. The supernatants (soluble proteins) were collected and used for Western blotting or LC-MS/MS analysis.

### 2.4. Sample Preparation for LC-MS/MS Analysis

CETSA samples were denatured in 8 M urea and reduced with 500 mM TCEP for 1 h at room temperature, followed by alkylation with 500 mM iodoacetamide for 1 h. The buffer was then changed to 200 mM triethylammonium bicarbonate using a 10 K centrifugal filter. Proteins were digested with MS-grade trypsin (1:40) at 37 °C for 16 h. The resulting peptides were labeled with a 6-plex TMT reagent (Thermo-Fisher Scientific, Waltham, MA, USA) according to the manufacturer’s protocols, and the reaction was quenched with 5% hydroxylamine. TMT-labeled samples (TMT-126: 55 °C DMSO; TMT-128: 55 °C UA; TMT-129: 55 °C DMSO; TMT-131: 60 °C UA) were pooled, vacuum-dried, dissolved in 0.5% formic acid, and desalted using a C18 macro spin column. After desalting, the samples were vacuum-dried and kept in a −80 °C freezer until HPLC fractionation.

### 2.5. High pH Reversed-Phase Liquid Chromatography for Peptide Fractionation

The dissolved 6-plex TMT-labeled sample was fractionated using an XBridge BEH Shield RP18 Column (130 Å, 2.5 µm, 4.6 × 150 mm, Waters) on NexeraXR HPLC (Shimadzu, Kyoto, Japan) with a 70 min gradient from 5% to 95% mobile phases B at a flow rate of 1.0 mL/min. Mobile phase A consisted of 5 mM ammonium formate in 100% water and mobile phase B consisted of 5 mM ammonium formate in 95% acetonitrile; both buffers were adjusted to a pH of 10 with ammonium hydroxide. A total of 40 fractions were collected using an FRC-10A fraction collector (Shimadzu, Kyoto, Japan) after the elution started with an interval of 1 min for each fraction. The 40 fractions were concentrated into 10 fractions. The concatenated fractions were dried and kept in a −80 °C freezer until LC-MS/MS analysis.

### 2.6. Liquid Chromatography–Tandem Mass Spectrometry

Ten fractions were analyzed using an LC-MS/MS system consisting of UltiMate U3000 RSLCnano (Thermo-Fisher Scientific, Waltham, MA, USA) and an Exploris 480 mass spectrometer with a nano-electrospray source, EASY-Spray Sources (Thermo-Fisher Scientific, Waltham, MA, USA). Peptides were first trapped in a precolumn (C18, 75 μm × 2 cm, nanoViper, Acclaim PepMap100, Thermo Fisher Scientific) and then applied to an analytical column (C18, 75 μm × 50 cm PepMap RSLC, Thermo Fisher Scientific) at a flow rate of 250 nL/min. The mobile phases were composed of 100% water (A) and 100% acetonitrile (B), each containing 0.1% formic acid. The LC gradient began with 5% B, maintained with 5% B over 8 min, ramped to 25% B over 101 min, followed by 50% B over 10 min, and increased to 95% B for 1 min, and then was held constant for 8 min, and ended with 5% B over 1 min. After a gradient, the column was re-equilibrated with 5% B for 10 min before the next run. The voltage applied to produce an electrospray was 1800 V. The Exploris 480 mass spectrometer operated in data-dependent mode, automatically switching between MS and MS/MS with a 2 s cycle time. Full scan MS spectra (400–1600 *m*/*z*) were acquired with an auto maximal injection time mode at a resolution of 120,000 and an automatic gain control (AGC) target value of 1.0 × 10^6^. MS/MS spectra were acquired from 110 *m*/*z* at a resolution of 30,000 with a high energy collision dissociation (HCD) of 38% normalized collision energy within a 1.2 Da isolation window. The AGC target value was 1.25 × 10^5^ with an auto maximal injection time mode. The exclusion time for previously fragmented ions was 60 s within 10 ppm.

### 2.7. Protein Identification and Quantitation

The Integrated Proteomics Pipeline using built-in search engines (IP2, version 6.5.5, Integrated Proteomics, San Diego, CA, USA) was utilized for data analysis with the UniProt human protein database (January 2023, reviewed 20,404 proteins). The reversed sequences of all proteins were appended into the database for calculation of the false discovery rate (FDR). ProLucid [21] was used to identify the peptides, a precursor mass error of 5 ppm, and a fragment ion mass error of 50 ppm. Trypsin was selected as the enzyme, with two potential missed cleavages. TMT modification (+229.1629) at the N-terminus and lysine residue by the labeling reagent and carbamidomethylation at cysteine were chosen as static modifications. Oxidation at methionine was chosen as variable modification. Reporter ions were extracted from small windows (±20 ppm) around their expected m/z in the HCD spectrum. The output data files were filtered and sorted to compose the protein list using the DTASelect (The Scripps Research Institute, La Jolla, CA, USA) with two or more peptide assignments for protein identification and a false positive rate less than 0.01 [22].

Quantitative analysis was performed using Census in the IP2 pipeline (Integrated Proteomics, San Diego, CA, USA) using only the unique peptides. The intensity at a reporter ion channel for a protein was calculated as the sum of the intensities of that reporter ion from all constituent peptides of the identified protein [23]. Reverse and potential contaminant proteins were removed. The protein intensities summed by the reporter ion intensities of all identified TMT-labeled peptides were uploaded to the Perseus platform (version 1.6.14.0). The data were normalized by subtracting the median values based on columns after being transformed into log_2_ values. Quantitative differences in protein levels (log_2_ fold change, FC) were then calculated as log_2_ [intensity of proteins in the UA sample] − log_2_ [intensity of proteins in the DMSO sample] at both 55 °C and 60 °C, and significant proteins were then selected.

### 2.8. In Silico Docking Study

Molecular docking analysis was performed using Discovery Studio 2018 software. The structure of the SERCA was obtained from the Protein Data Bank (PDB: 7E7S). For ligand docking, DOCKER (a grid-based molecular docking method using CHARM forcefield) was used. Binding sites were defined by receptor cavities. The ligand was docked to the binding site of the protein and the top 10 hits were generated. The binding energy (CDOCKER energy) was calculated.

### 2.9. ATPase Activity Assay

The ER Enrichment Kit (Invent Biotechnologies, Minneapolis, MN) was used to isolate ER proteins from liver cells (LX-2), according to the manufacturer’s instructions. The ER fraction was solubilized in 0.5% Triton X-100 buffer (50 mM Tris-HCl (pH 7.4), 150 mM NaCl, 0.5% Triton X-100, 1 mM EDTA, 1 mM EGTA). The ER protein at 1 μg/μL was preincubated with various concentrations of UA for 30 min, followed by measurement of ATPase activity. The ATPase activity was determined using an ATPase assay kit (Abcam, ab234055) according to the manufacturer’s instructions.

### 2.10. Lipid Droplet Staining

Cells were seeded in 12-well plates and co-treated with 0.5 mM PA and drugs for 24 h, followed by treatment with BODIPY 493/503. Images were captured using an LSM980 confocal microscope (Zeiss, Oberkochen, Germany) at 400× magnification. Green fluorescence intensity was quantified using Image J software (ver. 1.53) (National Institutes of Health, Bethesda, MD, USA).

### 2.11. Calcium Analysis

For mitochondrial or cytosolic calcium analysis, HepG2 cells were grown in 8-well chamber slides. Cells were co-treated with 0.5 mM PA and drugs for 24 h and then treated with either Rhod-2/AM (Invitrogen, Waltham, MA, USA, R1244), Mag-Fluo-4/AM (Invitrogen, Waltham, MA, USA, M14206), or Fluo-4/AM (Invitrogen, Waltham, MA, USA, F14201) for 30 min. Cells were washed with a Ca^2+^-free KRH buffer and live-imaged using an LSM980 confocal microscope (Zeiss, Oberkochen, Germany) at 400× magnification. Fluorescence signal intensity was quantified using Image J software (ver. 1.53) (National Institutes of Health, Bethesda, MD, USA).

### 2.12. Mitochondrial ROS Measurement

Mitochondria ROS levels were assessed using red fluorescent mitochondrial superoxide indicator MitoSOX (Invitrogen, Waltham, MA, USA, M36008). Cells were treated with drugs for 4 h, followed by incubation with 5 μM MitoSOX and Hoechst33342 for 10 min. The cells were then washed with PBS and live-imaged using an LSM980 confocal microscope (Zeiss, Oberkochen, Germany) at 400× magnification. The intensity of the red fluorescence was quantified using Image J software (ver. 1.53) (National Institutes of Health, Bethesda, MD, USA).

### 2.13. Transfection

Cells were transfected with the 100 nM SERCA siRNA (Bioneer, Daejeon, South Korea) for 24 h using Lipofectamine RNAiMAX reagent, according to the manufacturer’s instructions. A SERCA (FLAG-DYK-tagged) human ORF clone was used to generate the mutant SERCA vector. Cells were transfected with the SERCA plasmid using Lipofectamine 3000 reagent (Invitrogen, Waltham, MA, USA) according to the manufacturer’s instructions. To measure ER calcium, cells were transfected with CMV-ER-LAR-GECO1 (Addgene, Cambridge, MA, USA, #61244) for 48 h using Lipofectamine 3000 reagent, and the images were captured using an LCM980 confocal microscope (Zeiss, Oberkochen, Germany).

### 2.14. Materials

Urolithin A (SML1791), dimethyl sulfoxide (D2650), palmitic acid (P5585), Duolink^®^ In Situ Red Starter Kit (DUO92101), and Triton X-100 were purchased from Sigma-Aldrich (St. Louis, MO, USA). Rhod2-AM (R1244), Fluo4-AM (F14201), Mag-Fluo-4 AM (M14206), MitoSOX (M36008), Hoechst33342 (H3570), BODIPY 493/503 (D3922), Lipofectamine RNAimax (13778075), Lipofectamine 3000 (2757100), protease and phosphatase inhibitor solution (78441), DMEM (11995065), fetal bovine serum (FBS) (16000044), bovine serum albumin (A2153), and antibiotics were purchased from Thermo-Fisher Scientific (Waltham, MA, USA).

### 2.15. Statistical Analysis

Statistical analysis was performed using GraphPad Prism 9.0. All results are presented as the mean ± standard deviation (SD) values. Statistical significance was determined using Student’s *t*-tests (* *p* < 0.05, ** *p* < 0.01, *** *p* < 0.001, **** *p* < 0.0001, ns: not significant).

## 3. Results

### 3.1. Effects of UA on Palmitic Acid-Induced Cellular Stress

Palmitic acid (PA), a saturated fatty acid, induces ER stress when excessively accumulated in cells, which is associated with pathological conditions such as MAFLD [24]. We established a metabolic disease model by treating HepG2 with 0.5 mM PA for 24 h to induce lipid accumulation and ER stress. We then investigated whether UA could protect PA-treated cells. To determine the optimal concentration of UA, an MTT assay was performed, and the IC_50_ value at 72 h was determined to be approximately 40 µM (Figure 1A). Based on this, we found that 20 and 40 µM of UA increased both the proliferation (Figure 1B) and viability (Figure 1C) of PA-treated HepG2 cells in a dose-dependent manner. CHOP is a key transcription factor involved in ER stress-induced cell death. Using a cell model expressing the fluorescent reporter mCherry under the control of the CHOP gene promoter [25], we confirmed that UA reduced ER stress in PA-treated cells (Figure 1D). In addition, UA treatment reduced the elevated lipid levels induced by PA (Figure 1E). These results demonstrate that UA enhances cell survival in PA-treated HepG2 cells by alleviating cellular stress.

### 3.2. Target Identification of UA by CETSA-LC-MS/MS

To elucidate the mechanism by which UA reduces ER stress, we performed target identification using a cellular thermal shift assay (CETSA)-LC-MS/MS. CETSA is a method used to assess changes in the thermal stability of a protein caused by the binding of a compound to the protein without chemical labeling of the compound. HEK293 cells were treated with DMSO (control) or UA, heated at 55 °C and 60 °C, and the cells were lysed to obtain proteins. We performed TMT labeling on the CETSA samples to allow multiplex quantification, followed by mass spectrometry analysis (Figure 2A). A total of 5175 proteins were detected in our samples by MS analysis. Of these, 271 proteins exhibited more than 1.5-fold resistance to heat denaturation (log_2_ FC = (log_2_ [the intensity of proteins in the UA sample at 60 °C] − log_2_ [the intensity of proteins in the DMSO sample at 60 °C]) (Figure 2B). Analysis of the subcellular location of these candidate target proteins revealed that, interestingly, most of them were located in the ER or mitochondria (Figure 2B).

Through functional analysis of 17 ER proteins and 26 mitochondrial protein candidates, we identified that the target candidates are involved in calcium regulation, protein processing, phospholipid and lipid metabolism, energy metabolism, and oxidative reactions (Figure 2C). The ER is the largest calcium reservoir in the cell, and the calcium concentration within the ER is critical for various physiological activities and signal transduction. Because calcium imbalance increases cellular stress and affects cell survival, we focused on calcium-regulating proteins, SERCA (sarcoplasmic/endoplasmic reticulum calcium ATPase), ESYT1 (Extended Synaptotagmin 1), and CCDC47 (Coiled-Coil Domain Containing 47), when selecting candidate target proteins for UA. Since SERCA directly regulates ER calcium levels by pumping calcium into the ER, targeting SERCA could have a direct effect on ER calcium balance. Therefore, we considered SERCA as a primary target candidate for UA (Figure 2C).

### 3.3. Validation of Target Protein of UA

To validate the binding of UA and SERCA by CETSA analysis, UA-treated cells were exposed to heat in the range of 40 °C to 64 °C and the stability pattern was confirmed (Figure 3A). In addition, isothermal CETSA was performed with different concentrations of UA, using a constant heat of 52 °C (Figure 3B). Under these conditions, the EC_50_ value for UA binding to SERCA was approximately 26 µM (Figure 3B). We also performed a binding assay with CCDC47, one of the target candidates for UA, and obtained a higher EC_50_ value (approximately 38 µM) compared to SERCA.

To explore the binding site of UA on SERCA, an *in silico* docking simulation was performed (Figure 3E). When the binding of UA to the 3D structure of SERCA (PDB: 7E7S) was analyzed, the CDOCKER energy value, which represents the binding affinity of the ligand–protein complex, was −20.57 kcal/mol. SERCA is composed of three main domains; actuator domain, nucleotide domain, and phosphorylation domain [26]. The actuator domain is responsible for the structural changes in SERCA related to calcium ion transport [27], and the predicted binding pocket for UA is located within this domain (Figure 3D,E). Specifically, the oxygen of UA is expected to form a hydrogen bond with N201 and R198, while the ring structure of UA is predicted to form a Pi–cation interaction with K204 and K234 (Figure 3C). These results suggest that UA may modulate SERCA function by stabilizing critical domains that are involved in structural transitions.

In addition, to validate the predicted binding sites, we constructed mutant versions of the vector by substituting alanine for each site using a FLAG-tagged SERCA expression vector. HEK293 cells were then transfected with either wild-type (WT) or alanine mutant (R198A, K234A) vectors, followed by a CETSA. The significantly increased stability of SERCA by UA was only observed in cells transfected with WT SERCA vectors (Figure 3F), highlighting that R198 and K234 play critical roles in the binding of UA to SERCA. To determine whether UA acts as an agonist or antagonist on SERCA, we subsequently performed an ATPase activity assay with isolated ER proteins. ER proteins fractionated from liver cells were preincubated with UA, followed by the measurement of ATPase activity levels. Treatment with UA increased ATPase levels in a dose-dependent manner, whereas treatment with thapsigargin (TG, SERCA inhibitor) decreased ATPase levels (Figure 3G). Taken together, our *in vitro* and *in silico* assays support the idea that UA regulates SERCA function by binding directly to SERCA.

### 3.4. Modulation of Calcium Levels in Cellular Organelles by UA

To monitor changes in ER calcium levels upon UA treatment, HepG2 cells were transfected with a LAR-GECO1 vector, a fluorescent protein-based calcium indicator [28]. The cells were then treated with DMSO, UA, CDN11163 (CDN, a known agonist of SERCA) [29], and thapsigargin (TG, a known antagonist of SERCA) [30]. UA treatment significantly increased ER calcium levels, similar to CDN-treated cells, whereas TG treatment decreased ER calcium levels, suggesting that UA acts as a SERCA agonist to enhance calcium ion transport to the ER (Figure 4A). PA has been shown to cause hyperactivation of the ER calcium channel IP_3_R, leading to mitochondrial calcium overload and dysfunction [31]. In PA-treated cells, ER calcium levels decreased by more than 50%. However, UA treatment, similar to CDN treatment, restored ER calcium levels (Figure 4B, Appendix A). PA increased cytosolic calcium, but both UA and CDN treatments reduced the elevated cytosolic calcium (Figure 4C, Appendix A). PA treatment also increased mitochondrial calcium, which was subsequently reduced by UA (Figure 4D). These results suggest that UA alleviates the disrupted intracellular calcium homeostasis induced by PA.

### 3.5. UA-Mediated Regulation of Ca^2+^ Homeostasis and Its Role in ER Stress Reduction

To investigate whether the regulation of intracellular calcium homeostasis via SERCA is related to ER stress, we examined representative ER stress markers. PA treatment increased BiP at 3 h and elevated CHOP at 12 h (Figure 5A). The ER stress response is divided into three main pathways. We examined molecular markers related to the PERK-eIF2α pathway [32], which is involved in cellular stress responses and survival, after treatment with UA. PA increased BiP, the phosphorylation of PERK, and the phosphorylation of eIF2a at an early time point (6 h) (Figure 5B). Treatment with UA and CDN attenuated ER stress markers, whereas treatment with TG further increased them (Figure 5A). In addition, at the 24 h time point, PA treatment increased the levels of CHOP, a downstream marker of ER stress. Consistent with the previous results in Figure 1D, UA reduced the elevated CHOP levels (Figure 5C).

In addition, we investigated how UA-mediated regulation of calcium homeostasis in cellular organelles affects other cellular functions. UA reduced PA-induced mROS (Appendix A), which is consistent with the effects of UA to reduce the excess calcium accumulated in mitochondria by PA.

### 3.6. Influence of SERCA Knockdown on UA Activity

To determine whether the physiological effects of UA are mediated by interaction with its target protein, SERCA, we used siRNA to inhibit *SERCA* gene expression. After knocking down *SERCA*, we treated cells with PA and UA to determine whether the effect of UA was abolished. Similar to the previous results in Figure 4, when non-transfected cells were treated with PA and UA, there was an increase in ER calcium, a decrease in cytosolic calcium, and a decrease in mitochondrial calcium. However, in *SERCA*-depleted cells, there was no change in ER, cytosolic, and mitochondrial calcium despite treatment with UA (Figure 6A–C). This result confirms that UA is involved in calcium regulation within cell organelles by regulating SERCA. In addition, UA treatment did not affect BiP, p-PERK, and p-eIF2a levels in SERCA-deleted cells (Figure 6D), suggesting that UA contributes to alleviating ER stress by interacting with SERCA. Similarly, UA was unable to reduce mROS in *SERCA*-depleted cells (Appendix A). Our results strongly suggest that the physiological effects of UA on calcium homeostasis, ER stress reduction, and mitochondrial ROS levels are dependent on its interaction with SERCA, highlighting SERCA as a critical mediator of the protective effects of UA.

## 4. Discussion

Urolithin A (UA) is a natural compound metabolized by gut microbiota from ellagitannin and ellagic acid [11]. UA prevents the accumulation of damaged mitochondria by inducing mitophagy, enhances fatty acid oxidation and glycogen synthesis by activating the AMPK signaling pathway, and inhibits inflammatory responses by suppressing the NF-κB pathway [7,10,33]. Despite the diverse activities of UA, research on its target proteins is still poorly understood.

Palmitic acid (PA) treatment causes lipid accumulation and inflammation in the liver, similar to non-alcoholic fatty liver disease. In this study, we primarily observed the effect of UA in alleviating PA-induced cellular stress and protecting hepatocytes from lipid accumulation (Figure 1). Using CETSA-LC-MS/MS, a method for identifying target proteins using chemically unmodified compounds, SERCA was identified as a potential target of UA (Figure 2). In PA-treated liver cells, intracellular calcium homeostasis was disrupted, leading to ER stress; however, UA treatment alleviated this stress and restored calcium levels in the ER, cytoplasm, and mitochondria to their original state by regulating SERCA (Figure 4 and Figure 5). Recently, gene therapy aimed at increasing SERCA expression has been investigated for the treatment of heart failure [34]. This supports the importance of SERCA as a target in diseases where ER calcium regulation is impaired (Figure 7).

SERCA undergoes structural transitions between four major states (E1, E1∙Ca^2+^, E2∙Ca^2+^, and E2) that facilitate ATP hydrolysis and calcium ion transport [26]. Thapsigargin (TG), a non-competitive inhibitor of SERCA, binds to the E2 state, inhibiting ATPase activity and preventing calcium ion transport into the ER [35]. Crystal structures have shown that TG binds to SERCA at a pocket formed by the TM helices, with F256 identified as a key residue involved in this binding [36]. Our *in silico* docking analysis suggests that UA binds to the A (actuator) domain of SERCA, providing a new small molecule tool with a binding site distinct from TG. However, our current studies have not yet confirmed whether UA induces structural changes in SERCA. Investigation of potential structural changes caused by UA binding remains for future research.

Recently, mitochondria-associated ER membranes (MAMs) have been shown to play an important role in maintaining cellular calcium homeostasis and regulating metabolic processes [37]. Lee et al. revealed a mechanism by which UA disrupts the contact between the ER and mitochondria by suppressing the expression of the TGM2 protein in the MAM, thereby inhibiting calcium influx into the mitochondria and subsequently reducing mROS levels [38]. In line with this, we investigated the interactions between the ER and mitochondria in our HepG2 metabolic disorder model. The PLA assay between IP_3_R and VDAC1 showed that these two organelles are closer together in PA-treated cells and that this interaction is reduced upon UA treatment (Appendix A). UA’s ability to regulate organelle interactions helps reduce oxidative stress by preventing excessive calcium influx into mitochondria, making it a promising option for treating metabolic disorders. However, further research is needed to elucidate the detailed mechanisms by which UA regulates ER–mitochondria interactions and maintains cellular homeostasis.

Consistent with a previous study that found that UA reduces lipid accumulation [39], we observed that UA reduced lipid droplets increased by PA. Future research should investigate how the reduction in ER stress by UA is related to lipid levels, whether UA inhibits lipid uptake, and whether it can promote the degradation of lipid droplets via autophagy. This will help us better understand the mechanisms of UA regulation of cellular homeostasis and provide insight into the treatment of related diseases.

While further research is needed to fully understand the mechanism of UA, our study is significant because it provides the first evidence that UA acts as an activator of SERCA. This finding highlights the potential role of UA in protecting against ER stress-induced metabolic dysfunction.

## 5. Conclusions

This is the first study to identify UA as an activator of SERCA, which is responsible for organelle calcium regulation activity in hepatocytes undergoing ER stress by PA. Furthermore, we demonstrated that SERCA, a direct binding partner of UA, plays a critical role in the regulation of calcium flux in MAMs. Therefore, these results highlight the potential of UA in the treatment of ER stress-induced metabolic dysfunction.

## Figures and Tables

**Figure 1 biomolecules-14-01505-f001:**
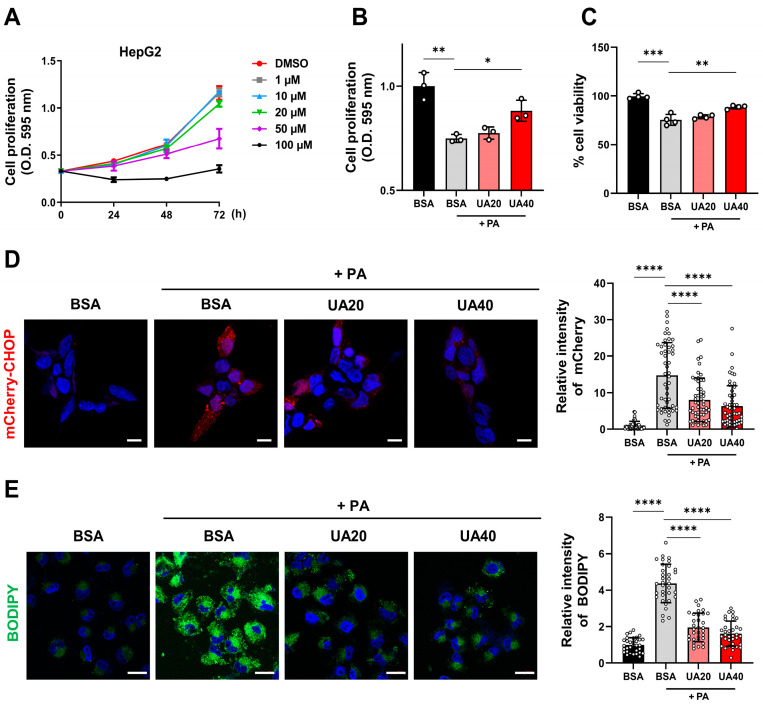
Effects of UA on cellular response, ER stress, and lipid accumulation in PA-treated hepatocytes. (**A**) The MTT assay was performed on HepG2 cells treated with various concentrations of UA (1–100 μM) for 24, 48, and 72 h to evaluate cell proliferation. (**B**) The MTT assay was used to measure the cell proliferation of HepG2 cells treated with PA (0.5 mM) and UA (20, 40 μM) for 24 h. (**C**) The Trypan blue exclusion assay was performed to evaluate the cell viability of HepG2 cells treated with PA (0.5 mM) and UA (20, 40 μM) for 24 h. (**D**) mCherry-CHOP stable HEK293 cells were treated with PA (0.5 mM) and UA (20, 40 μM) for 24 h, and mCherry fluorescence signals were measured. The expression of CHOP was visualized as a fluorescence signal and observed using confocal microscopy (scale bar: 10 μM). (**E**) HepG2 cells were treated with PA (0.5 mM) and UA (20, 40 μM) for 24 h, followed by staining of intracellular lipid droplets with BODIPY and observation via confocal microscopy (scale bar: 20 μM) (* *p* < 0.05, ** *p* < 0.01, *** *p* < 0.001, **** *p* < 0.0001).

**Figure 2 biomolecules-14-01505-f002:**
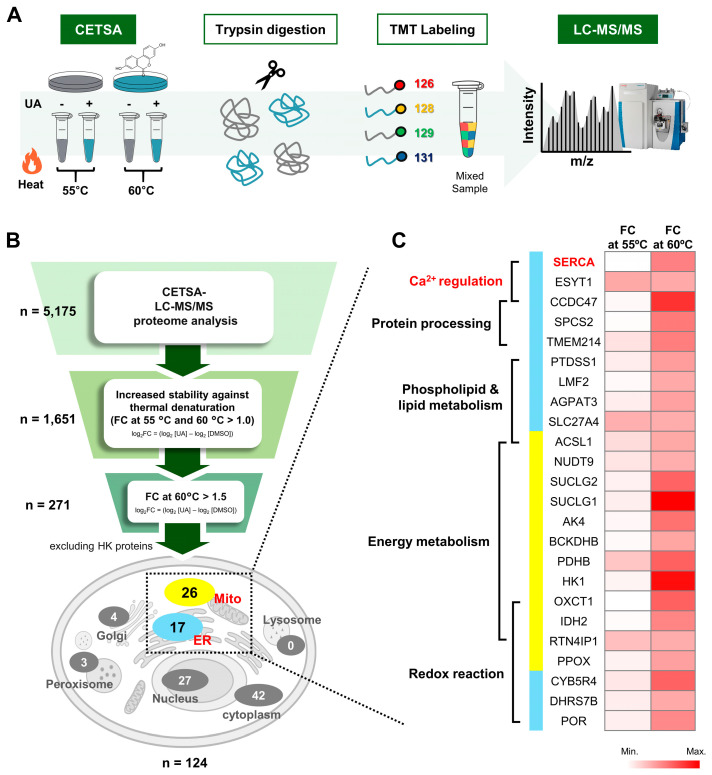
CETSA-LC-MS/MS for the identification of the target protein of UA. (**A**) Overview of the CETSA-LC-MS/MS method. HEK293 cells were treated with DMSO (control) or UA (20 μM) and subjected to thermal treatment (55 °C or 60 °C). Proteins were then extracted, digested with trypsin, and labeled with TMT reagents. The labeled peptides were analyzed by HPLC and LC-MS/MS. (**B**) Schematic diagram of the target selection identification criteria of the UA process. (**C**) Heatmap of target candidates of UA localized in the ER and mitochondria. Proteins are clustered by their functions, as indicated on the left side of the heatmap.

**Figure 3 biomolecules-14-01505-f003:**
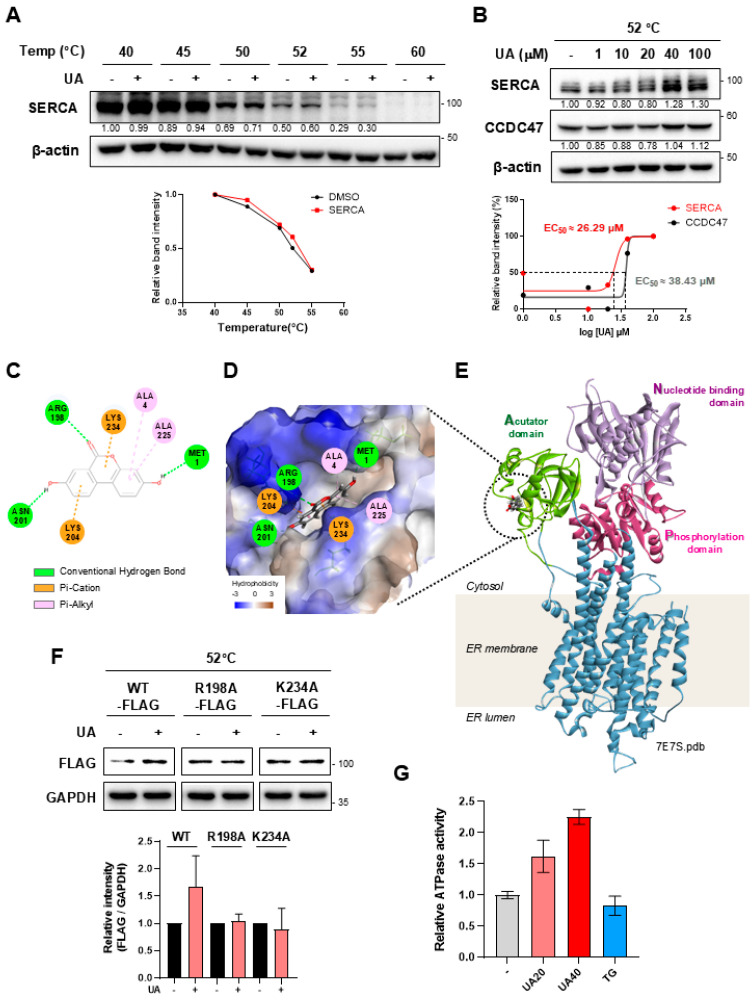
Validation of UA binding to SERCA. (**A**) Western blot analysis to evaluate the thermal stability of SERCA in HepG2 cells treated with UA (40 μM). (**B**) Comparison of the binding affinity between UA and two calcium-regulating target proteins: SERCA and CCDC47. HepG2 cells were treated with various concentrations of UA (1–100 μM), followed by isothermal CETSA at 52 °C. (**C**) The 2D diagram represents amino acids involved in UA binding to SERCA. (**D**) Predicted binding of UA to the actuator domain of SERCA, as visualized using Discovery Studio software 2018 (CDOCKER energy: −20.57 kcal/mol). (**E**) The 3D structure of the full-length SERCA protein (PDB: 7E7S), highlighting its four domains. (**F**) HEK293 cells were transfected with FLAG-SERCA(WT), FLAG-SERCA(R198A), or FLAG-SERCA(K234A) for 48 h and then treated with UA (40 μM). After heat treatment at 52 °C for 3 min, proteins were extracted and analyzed using a Western blot. (**G**) Measurement of ATPase activity of ER proteins extracted from LX2 cells. UA (20, 40 μM) and thapsigargin (0.1 μM) were each treated for 30 min before ATPase activity was assessed.

**Figure 4 biomolecules-14-01505-f004:**
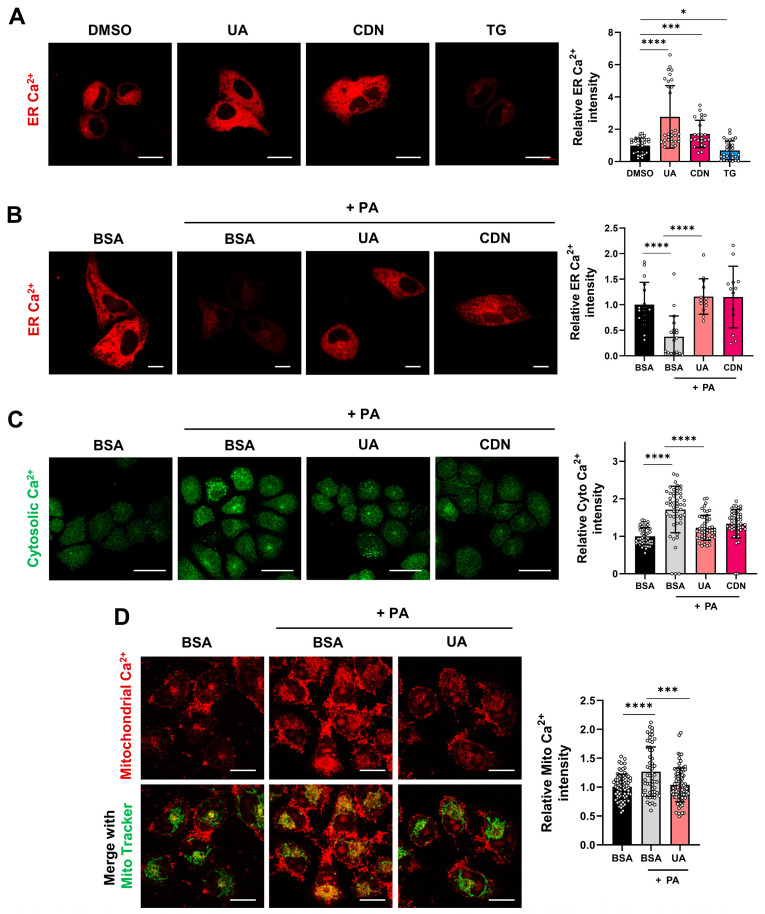
Intracellular calcium levels in PA-treated HepG2 cells with UA. (**A**) HepG2 cells were transfected with the ER calcium indicator ER-LAR-GECO vector for 48 h, followed by treatment with UA (40 μM), CDN1163 (10 μM), or TG (0.1 μM) for 6 h (scale bar: 20 μm). (**B**) ER calcium levels were measured in HepG2 cells treated with PA (500 μM) for 6 h, either alone or co-treated with UA (40 μM) and CDN1163 (10 μM) (scale bar: 10 μm). (**C**) Cytosolic calcium levels were assessed using the Fluo-4-AM after 6 h of treatment with PA, either alone or co-treated with UA (40 μM) and CDN1163 (10 μM) (scale bar: 20 μm). (**D**) Mitochondrial calcium levels were determined using the Rhod-2-AM after 24 h of treatment with PA, either alone or co-treated with UA (40 μM) and CDN1163 (10 μM) (scale bar: 20 μm) (* *p* < 0.05, *** *p* < 0.001, **** *p* < 0.0001).

**Figure 5 biomolecules-14-01505-f005:**
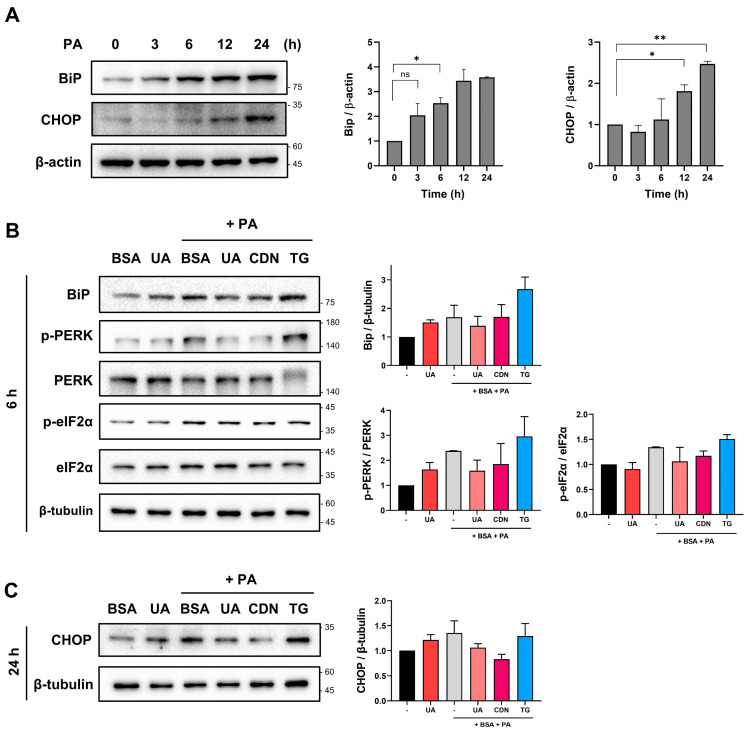
ER stress markers in PA-treated HepG2 cells with UA. (**A**) Western blot results showing changes in ER stress markers in HepG2 cells treated with PA. (**B**) HepG2 cells were treated with PA for 6 h in the absence or presence of UA (40 μM), CDN1163 (10 μM), and thapsigargin (0.1 μM). UA down-regulated the level of ER stress-related proteins. (**C**) HepG2 cells were treated with PA for 24 h in the absence or presence of UA (40μM), CDN1163 (10 μM), and thapsigargin (0.1 μM). UA down-regulated the level of CHOP (* *p* < 0.05, ** *p* < 0.01, ns: not significant).

**Figure 6 biomolecules-14-01505-f006:**
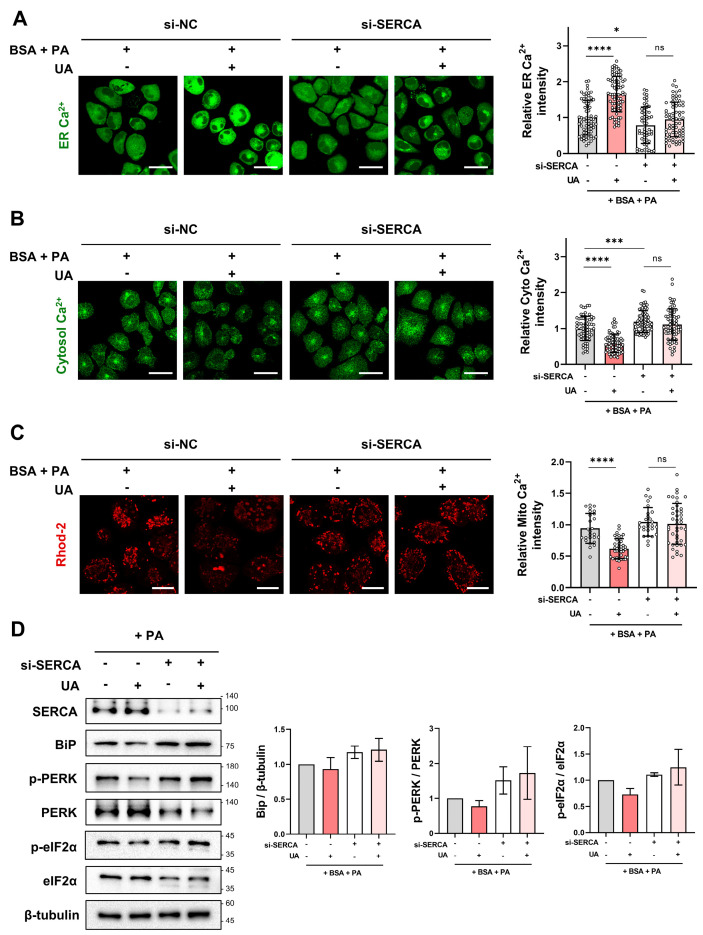
Impact of *SERCA* knockdown on UA activity in PA-treated cells. (**A**–**C**) Following *SERCA* knockdown with si-SERCA for 24 h, cells were co-treated with PA and UA (40 μM) for 6 h. ER, cytosolic, and mitochondrial calcium levels were then measured using Mag-Fluo-4 AM, Fluo-4 AM, and Rhod-2 AM, respectively (scale bar: 40 μm). (**D**) *SERCA* knockdown using si-*SERCA* for 24 h confirmed the effect of UA (40 μM) on PA-induced ER stress protein levels, as shown by Western blot (* *p* < 0.05, *** *p* < 0.001, **** *p* < 0.0001, ns: not significant).

**Figure 7 biomolecules-14-01505-f007:**
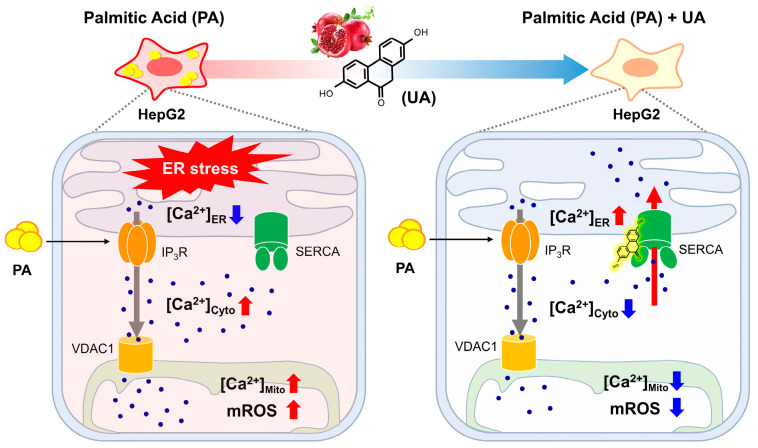
Schematic summary of the target proteins and mechanisms of action of UA. HepG2 cells stimulated with PA release ER calcium through IP_3_R, leading to ER stress. UA binds to SERCA, the ER calcium pump, replenishing ER calcium levels and maintaining calcium homeostasis. This mechanism helps protect the cells from stress-induced damage.

## Data Availability

The raw data and search results are available at MassIVE: MSV000096011.

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
