# Peer review of "Urolithin A Protects Hepatocytes from Palmitic Acid-Induced ER Stress by Regulating Calcium Homeostasis in the MAM"

_biomolecules, 2024, doi:10.3390/biom14121505_

Round 1

Reviewer 1 Report

Comments and Suggestions for Authors

Title: please do not use abreviations in the title-or shortly extend

Abstract: please use the commonly accepted format in aim/methods/results/conclusion form; in this manner, please try to separate results from conclusions (e.g. " In addition, UA inhibits PA-induced mitochondrial 24 ROS and lipid accumulation, further supporting its protective role")

Manuscript:

Materials: 2.1 chapter- please explain each substance, or add at the end of material- the enumeration of all the substances used are far too early in manuscript, as no method is yet described.

The method is properly described

Results: please do not use a conclusion as a title of a figure/table (e.g. figure 1) it should be a conclusion.

As well- a conclusion as a sub-chapter title- (e.g. : "3.4. UA regulates calcium levels within cellular organelles")

A title should only describe the results/not the conclusion (e.g.: "UA and calcium levels within cellular organelles") . The same in a figure/table.

The method, however, is extensively and properly described.

Discussion  chapter- too short for so many subchapters. Each subchapter in the results chapter should have a discussion subchapter- or, if not possible, grouping the results should emphasize a certain result.

Conclusion: there is no use of figures in the conclusions chapter which should be very concise. Please use the figure, if really needed, in the discussions.

English language- fine, no major issues.Novelty: Is the question original and well-defined? Do the results provide an advancement of the current knowledge?

Significance: good hypothesis, good results

Quality: high quality of an original article. 

Scientific Soundness: good- but for those closely related to the area of research.

Interest to the Readers: good soundness but for a limited area of research

Overall Merit: a well written article

English Level: Is the English language appropriate and understandable?

Author Response

Responses to the reviewers' comments

(Reviewer #1)

Major comments

  • Abstract: please use the commonly accepted format in aim/methods/results/conclusion form; in this manner, please try to separate results from conclusions (e.g. " In addition, UA inhibits PA-induced mitochondrial ROS and lipid accumulation, further supporting its protective role")

Response: Thank you for your valuable comment. We agree with your point and have removed this sentence to ensure the Abstract is more concise and clear accordingly (P1, L24-25).

  • Manuscript:

Materials: 2.1 chapter- please explain each substance, or add at the end of material- the enumeration of all the substances used are far too early in manuscript, as no method is yet described.

Response: Thank you for your constructive suggestions regarding formatting improvements. To enhance readers' understanding of the Materials section, we have adjusted the manuscript to list materials after the Methods section (P5, L219-225).

  • Results: please do not use a conclusion as a title of a figure/table (e.g. figure 1) it should be a conclusion. As well- a conclusion as a sub-chapter title- (e.g. : "3.4. UA regulates calcium levels within cellular organelles"). A title should only describe the results/not the conclusion (e.g.: "UA and calcium levels within cellular organelles") . The same in a figure/table. The method, however, is extensively and properly described.

Response: Thank you for your valuable comments on the manuscript. We have revised all titles and subtitles to ensure they objectively describe the experimental results without concluding statements accordingly (P5. L233; P6, L262; P11, L347; P12, L346: P14, L367).

  • Discussion chapter- too short for so many subchapters. Each subchapter in the results chapter should have a discussion subchapter- or, if not possible, grouping the results should emphasize a certain result.

Response: Thank you for highlighting this important aspect. We recognize that the discussion section was relatively short compared to the results, which led to some ambiguity in the presentation of each finding. In response to your valuable comments, we have revised the section by discussing and highlighting for each result, to ensure that the corresponding findings are clearly highlighted (P14, 377-384).

  • Conclusion: there is no use of figures in the conclusions chapter which should be very concise. Please use the figure, if really needed, in the discussions.

Response: Thank you for your clear suggestion. The summary figure is an important element that encapsulates the overall content of the manuscript and helps to effectively convey the main findings to the readers. Following your recommendation, we have moved this figure to the Discussion section.

Reviewer 2 Report

Comments and Suggestions for Authors

The goal of the present proposal is to identify a novel target protein of UA and elucidate its mechanism for alleviating palmitic acid (PA)-induced ER stress. Although, the current study is interesting and writing. I recommend the publication for this study after major revisions as the following, 

  1. Please provide densitometric analysis for Western blot figure 3A and B 

  1. Why did authors use cancer cell line HepG2 and not normal cell line (major point)? Please provide an answer with confirmation of this mechanism in normal cell line. 

  1. Could the authors add reference for the modern nomenclature for MAFLD? 

Reference 

A global survey on the use of the international classification of diseases codes for metabolic dysfunction-associated fatty liver I disease. Hepatol Int. 2024 Aug;18(4):1178-1201. doi: 10.1007/s12072-024-10702-5. 

Author Response

(Reviewer #2)

  • Please provide densitometric analysis for Western blot figure 3A and B 

Response: Thank you for suggesting this improvement. To improve the clarity of data interpretation, we have included the quantitative values for band intensity in the Western blot for Figure 3A and B, providing a clearer presentation of the densitometric analysis.

  • Why did authors use cancer cell line HepG2 and not normal cell line (major point)? Please provide an answer with confirmation of this mechanism in normal cell line. 

Response: Thank you for your insightful comment regarding the choice of cell line. HepG2 cells were chosen because they are widely used in 2D in vitro modeling of MAFLD due to their consistency in exhibiting key pathological features, including lipid accumulation and oxidative stress responses when treated with palmitic acid (PA) [1,2]. Compared to primary human hepatocytes (PHH), HepG2 cells offer greater consistency and robustness in inducing these MAFLD-related responses under PA treatment.

While PHH closely mimic the physiological liver environment, they are limited by rapid dedifferentiation, short culture durations, and higher fatty acid requirements for lipid accumulation [3]. In addition, PHH have a natural tendency toward lipid catabolism, making it difficult to model the consistent lipid accumulation and metabolic imbalance typical of MAFLD pathology [4].

In PHH, PA treatment can still induce ER stress and mitochondrial ROS (mROS), which could potentially be alleviated by UA treatment. However, given the inherent lipid catabolic activity of PHH, the differential recovery between UA-treated and untreated cells may be less pronounced than in HepG2 cells.

  1. Müller, F.A.; Sturla, S.J. Human in vitro models of nonalcoholic fatty liver disease. Current Opinion in Toxicology 2019, 16, 9-16.
  2. Minami, Y.; Hoshino, A.; Higuchi, Y.; Hamaguchi, M.; Kaneko, Y.; Kirita, Y.; Taminishi, S.; Nishiji, T.; Taruno, A.; Fukui, M.; et al. Liver lipophagy ameliorates nonalcoholic steatohepatitis through extracellular lipid secretion. Nature Communications 2023, 14, 4084.
  3. Wang, S.X.; Yan, J.S.; Chan, Y.S. Advancements in MAFLD Modeling with Human Cell and Organoid Models. Int J Mol Sci 2022, 23.
  4. Soret, P.A.; Magusto, J.; Housset, C.; Gautheron, J. In Vitro and In Vivo Models of Non-Alcoholic Fatty Liver Disease: A Critical Appraisal. J Clin Med 2020, 10.

3) Could the authors add reference for the modern nomenclature for MAFLD? 

Reference 

A global survey on the use of the international classification of diseases codes for metabolic dysfunction-associated fatty liver I disease. Hepatol Int. 2024 Aug;18(4):1178-1201. doi: 10.1007/s12072-024-10702-5. 

Response: Thank you for this helpful suggestion. We have included the suggested reference to emphasize the modern nomenclature of MAFLD.

Round 2

Reviewer 1 Report

Comments and Suggestions for Authors

improved

Reviewer 2 Report

Comments and Suggestions for Authors

The authors have successfully addressed all comments